# Characteristics and Kinetics of Bainite Transformation Behaviour in a High-Silicon Medium-Carbon Steel above and below the M_s_ Temperature

**DOI:** 10.3390/ma15020539

**Published:** 2022-01-11

**Authors:** Shima Pashangeh, Seyed Sadegh Ghasemi Banadkouki, Mahesh Somani, Jukka Kömi

**Affiliations:** 1Department of Mining and Metallurgical Engineering, Yazd University, University Blvd, Safayieh, Yazd P.O. Box 98195-741, Iran; pashangeh.a@gmail.com; 2Materials and Mechanical Engineering, Centre for Advanced Steels Research, University of Oulu, P.O. Box 4200, 90014 Oulu, Finland; mahesh.somani@oulu.fi (M.S.); jukka.komi@oulu.fi (J.K.)

**Keywords:** AHSS steel, dilatometric analysis, bainitic transformation, transformation kinetics, martensite

## Abstract

This work deals with the kinetic aspects of bainite formation during isothermal holding above and below the martensite start (M_s_~275 °C) temperature using a low-alloy, high-silicon DIN 1.5025 steel in a range suitable for achieving ultrafine/nanostructured bainite. Dilatation measurements were conducted to study transformation behaviour and kinetics, while the microstructural features were examined using laser scanning confocal microscopy and electron backscatter diffraction (EBSD) techniques combined with hardness measurements. The results showed that for isothermal holding above the M_s_ temperature, the maximum bainitic transformation rate decreased with the decrease in isothermal holding temperature between 450 and 300 °C. On the other hand, for isothermal holding below the M_s_ temperature at 250 and 200 °C, the maximum rate of transformation was achieved corresponding to region I due to the partitioning of carbon and also possibly because of the ledged growth of isothermal martensite soon after the start of isothermal holding. In addition, a second peak was obvious at about 100 and 500 s, respectively, during holding at 250 and 200 °C due to the occurrence of bainitic transformation, marking the beginning of region II.

## 1. Introduction

In the last few decades, advanced high-strength steels (AHSSs) with multiphase microstructures, essentially comprising fine phase mixtures of bainite (B), martensite (M) and retained austenite (RA), have attracted renewed interest due to their superb combinations of high strength and good ductility as well as high strain hardening capacity, and are being considered as potential candidates for use in automobile and industrial applications [1,2,3]. Steels processed at a temperature close to M_s_ temperature via the quenching and partitioning (Q&P) route with essentially finely divided martensite–austenite–nanostructured bainite structures, as well as quenching and bainititizing (Q&B) treatment with mainly ultrafine/nanostructured bainite–austenite structures, are two such groups of multiphase third-generation AHSSs that show greatly improved mechanical property combinations, including good ductility imparted by transformation-induced plasticity (TRIP), an effect of the finely divided retained austenite (RA) in the steels [2,4,5]. In order to achieve a mixture of ultrafine bainite and finely distributed, carbon-enriched retained austenite, isothermal heat treatment procedures close to (both above and below) the M_s_ temperature have been suggested [6,7]. These isothermal heat treatments have been shown to impart excellent mechanical properties to the steel, including ultrahigh strength, adequate toughness, reasonable ductility and good wear resistance [7,8].

In general, AHSSs contain bainite as an important strengthening constituent with or without the presence of an RA phase as an interspersed minor constituent. Customarily, the bainite microstructure has been broadly classified into upper and lower bainites, due to the difference in distribution of carbide precipitates. In upper bainite, most of the carbides are normally distributed within the interfaces of the ferritic sheaves, whereas in lower bainite, carbides are known to form inside the ferritic platelets without significant transfer of carbon to the austenite in the interfaces [9,10]. In contrast, the quenching and partitioning (Q&P) process has been proposed as a potential means of improving the balance of elongation to fracture and tensile strength for AHSSs, wherein the steel is austenitised, quenched to a temperature between the M_s_ and M_f_ temperatures and held at an appropriate temperature for a proper time to enable partitioning of carbon from supersaturated martensite to untransformed austenite, which can then be partially or fully stabilized on cooling to room temperature [11,12]. 

In the achievement of a multiphase microstructure, including a desired bainite fraction, by isothermal holding around the M_s_ temperature, the previous study clearly showed that selection of austempering temperature was more effective than manipulating holding time in the course of isothermal heat treatment [13]. To design the control of the heat treatment, prior knowledge of phase transformation kinetics is of paramount significance. Among the existing computational models that are used to calculate the decomposition fraction of the austenite phase, the Kirkaldy–Venugopalan model, e.g., [14], and the JMAK type (Johnson–Mehl–Avrami–Kolmogorov) model, are quite common, e.g., [15,16,17]. Based on the JMAK equation combined with the additivity rule, some mathematical models were developed using the phase transformation data and proposed for different multiphase steels, including dual-phase (DP), TRIP and complex-phase (CP) steels [18,19,20]. For heat treatments promoting the formation of phase mixtures, including bainite and martensite, it is imperative that we be able to model the process, particularly when all the austenite phase may not transform into bainite. Therefore, the JMAK equation in differential form [16,21] still remains a useful and practical tool that includes a description of the maximum bainite fraction transformed at different temperatures. 

Previous investigations have shown that the characteristic features of different phases, including the amount, size, morphology and orientation, are important factors that influence the final mechanical properties of bainitic steels [22,23]. Both the carbon content as well as austempering temperature were important parameters influencing the kinetics of phase transformation [24,25]. As the theory of T_0_ curve suggests [26], in high-C bainitic steels, more supercooled, carbon-enriched, untransformed austenite will be realized following isothermal heat treatment at an austempering temperature, as the incompleteness of bainitic phase transformation increases. Besides, more carbon atoms are available in high-carbon steels to partition from the transformed bainitic ferrite to the remaining untransformed austenite and, accordingly, the thermal stability of austenite increases. Consequently, more austenite will be retained at room temperature, mostly in large blocks depending on the experimental parameters [6,22,27].

In previous investigations, researchers have correlated the mechanical properties of AHSSs with the characteristics and fractions of bainitic microstructures obtained after different isothermal holding treatments above and below the M_s_ [28,29,30]. However, only a few researchers studied thoroughly the kinetics of bainitic phase transformation in low-alloyed, high-silicon, medium-carbon steels, which not only clarifies the importance of various mechanisms operating during phase transformation under different heat-treatment conditions, but also allows the prediction of final microstructures and properties of heat-treated low-alloy steel and helps with the design of the process. Therefore, this study presents the characteristics and kinetics of ultrafine (nanostructured) bainite transformation during isothermal holding for 1 h at different isothermal temperatures in the vicinity of M_s_ temperature. The bainitic transformation behaviour and kinetics were characterized using the dilatation measurements made with a Gleeble thermomechanical simulator using a JMAK-type of analysis [31]. The results were further corroborated by Vickers macrohardness measurements and detailed microstructural investigations of multiphase microstructures conducted using laser scanning confocal microscopy, EBSD characterization and XRD measurements.

## 2. Materials and Methods

### 2.1. Experimental Material

A high-silicon, medium-carbon steel sheet (DIN1.5025 grade) of 1 mm thickness with the chemical composition (all concentrations in wt.%) Fe-0.529C-1.670Si-0.720Mn-0.120Cr was used in this research. A high amount of Si is beneficial because of its influence in preventing (or at least delaying) carbide formation and/or its growth during the phase transformation and finally promoting carbon partitioning to the adjacent remaining austenite phase. However, the Mn content (0.72 wt.%) was lower than normally desired to promote the stabilization of an austenite phase during carbon partitioning.

### 2.2. Heat-Treatment Cycles

Suitable heat-treatment schedules above and below the M_s_ temperature were planned for conducting typical quenching and bainitizing (Q&B) and Q&P processes, respectively, in a thermomechanical simulator (Gleeble 3800 (New York, NY, USA)) to investigate the transformation behaviour and/or evolving microstructural mechanisms. The determination of critical temperatures, viz., the start (A_c1_) and finish (A_c3_) austenite formation temperatures, M_s_ temperature, as well as the bainite start temperature (B_s_), were therefore considered necessary to design proper heat-treatment processes. Accordingly, A_c1_ (765 °C), A_c3_ (835 °C) and M_s_ (275 °C) temperatures were determined via dilatometry measurements by choosing the heating and cooling rates of 0.2 °C/s and 150 °C/s, respectively (detailed discussion presented elsewhere [32]), while the B_s_ temperature (471 °C) was estimated using the empirical equation proposed in [33]. These measurements enabled the design of experimental cycles for Q&B and Q&P heat treatments on the thermomechanical simulator using specimens of dimensions 30 × 9 × 1 mm^3^, resulting in the formation of a uniform central hot zone of ~4 mm width in the specimens. 

Various steps in heat-treatment cycles included heating to the austenitization temperature of 900 °C at 50 °C/s, holding for 5 min, then cooling at 50 °C/s to different temperatures for bainitization treatment between 450 and 200 °C in steps of 50 °C and holding for 1 h at the selected temperatures. This was followed by cooling to room temperature at 50 °C/s (schematically shown in Figure 1). The isothermal holding temperatures selected for Q&B experiments above the M_s_ temperature were 450, 400, 350 and 300 °C, and those for Q&P experiments were 250 and 200 °C below the M_s_. Quenching to 250 and 200 °C facilitated initial martensite fractions of about 20 and 57%, respectively. The martensite fractions on quenching to different temperatures (250 and 200 °C) were estimated from the dilatation curves using the lever rule and more details are given in our previous work [5]. Dilatation measurements were performed for all steps during the quenching and isothermal holding processes, as well as final cooling to room temperature.

### 2.3. Microstructural Observation and Hardness Measurement

In order to reveal the microstructural details of the Gleeble-simulated specimens, the specimens were prepared according to the ASTM E3 standard and etched with a 2% nital solution. An initial investigation into microstructure was made using a 3D laser scanning confocal microscope (model Keyence VK-X200 (Itasca, IL 60143, USA)). The specimen preparation for electron backscatter diffraction (EBSD) involved a final step of polishing with colloidal silica (0.04 μm) suspension in methanol. Later, the specimens were subjected to detailed microstructural investigation using the EBSD facility equipped with a Zeiss Ultra Plus field emission scanning electron microscope (FESEM (Zeiss, Germany)). The EBSD analysis was performed with the help of HKL Channel 5 system software. Select specimens were subjected to thorough metallographic examination using a 200 kV Jeol JEM-2200FS (Peabody, MA, USA) transmission electron microscope (TEM). Thin slices (100 μm) of samples were cut to prepare the thin foils for the TEM study. The samples were thinned up to 50 μm by mechanical polishing and 3 mm discs were punched to conduct twin-jet electropolishing in an electrolyte composed of 10% perchloric acid and 90% acetic acid, maintained at 10 °C.

For conducting hardness measurements, the samples were polished to mirror-finish flat surfaces. Vickers hardness tests were conducted on all heat-treated samples using a load of 30 kg. An average of five different measurements was reported for the hardness evaluation.

### 2.4. Microphase Analysis

The presence of different phases in the specimens was determined using X-ray diffraction (XRD) analysis (Rigaku SmartLab 9 kW) (Rigaku Corp., Tokyo, Japan). The measurements were made using a CoK_α_ radiation source at 135 mA and 40 kV conditions in 2θ range between 45 to 130° and the rotation was performed at 7.2°/min. The volume fraction and also the lattice parameter of retained austenite were calculated using a direct comparison method, thus comparing the integrated intensities of diffracted planes for FCC (face-centred cubic), including (111), (200), (220) and (311) planes and diffracted planes of BCC (body-centred cubic), including (101), (002), (112) and (202) planes, respectively. Then, the carbon contents were determined according to the following equation [34]: a_γ_ [^o^A] = 3.572 + 0.033x_C_ + 0.0012x_Mn_ + 0.0056x_Al_ + 0.00157x_Si_(1)
where a_γ_ is the austenite phase lattice parameter in angstroms and x_C_, x_Mn_, x_Al_, x_Si_ are the mass fractions of carbon, manganese, aluminium and silicon contents (in wt.%), respectively.

### 2.5. Kinetics Data Collection 

The Avrami-type function (Equation (2)) [35] was used to describe the relationship between the progress of bainite formation with a specific holding time at each isothermal holding temperature.
x = 1 − exp(−kt^n^)(2)
where x is the phase fraction which transformed after a specific time t, n is the Avrami exponent and k is a temperature-dependent rate coefficient of the transformation reaction. The Avrami exponent n depends on the nucleation mechanism and growth of the bainite phase. Equation (1) can be reorganized as given below in Equation (3) in order to be able to determine the k and n values: ln [−ln (1 − x)] = lnk + nlnt(3)

The values of k and n were achieved by plotting the organized “Avrami” relationship. The k and n values were the intercepts of the best fit lines on the *y*-axis and the slopes of the corresponding regression equations, respectively. 

## 3. Results

### 3.1. Progress of Bainite Formation

Figure 2 depicts the dilatation temperature measurements made on Q&B and Q&P specimens quenched and held both above and below the M_s_ temperature in the range 450–200 °C for 1 h. The small scatter in the initial slope during cooling (Figure 2) may be due to minor variation in the course of dilatometry tests. The curves in Figure 2a show a percent change in width (contraction) during cooling from 900 °C to a particular temperature above the M_s_, then expansion during subsequent isothermal holding at these temperatures as a result of conventional bainite transformation. Figure 2b shows the dilatation measurements made on Q&P specimens held at 250 and 200 °C for 1 h. Prior to holding, the occurrence of dilatation as a consequence of the decrease in temperature below M_s_ resulted in an appreciable expansion, obviously due to the athermal martensite formation giving the initial martensite fractions of about 20 and 57% at 250 and 200 °C, respectively. In addition, referring to the plots shown in Figure 2b, the expansion for the Q&P specimen held at 250 °C is several times greater than the dilatation that occurred in athermal martensite formation, revealing that the austenite-to-bainite transformation continues during isothermal holding. Prior martensite is known to accelerate the subsequent bainitic transformation [36,37]. So, the presence of initial martensite precisely after quenching to 250 and 200 °C (as marked in Figure 2b) might have accelerated subsequent bainite transformation and caused a noticeable expansion that was significantly higher than that predicted for partitioning of carbon, as can be discerned from the dilatation curves. 

Figure 3 reveals the dilatation measurements conducted on Q&B and Q&P specimens held for 1 h at different temperatures, both above and below the M_s_ temperature, respectively. To assist interpretation, both logarithmic (Figure 3a,c) and linear (Figure 3b,d) time scales have been used. Referring to Q&B heat-treatment conditions (Figure 3a,b), the occurrence of bainite transformation during isothermal holding can be expediently described by Avrami-type functions (typical S-type curves on logarithmic time scales; Figure 3a), as the transformations appeared near completion at about 1 h regardless of maximum bainite fractions that formed at different isothermal holding temperatures between 450 and 300 °C. Moreover, the dilatation curves of samples isothermally held at 450 and 400 °C displayed contraction beyond about 100 and 470 s, which can be attributed to the occurrence of extensive tempering of martensitic/bainitic laths, besides the possible carbide formation, despite the high silicon content [29]. A quick comparison of the dilatation curves in the temperature range of 300–450 °C (Figure 3a) showed that the time at which the bainitic transformation started increased continuously with the decreasing isothermal holding temperature. The shape of the initial part (up to about 10 s) of the dilation curve at 450 °C, however, is unreliable and is attributed to the error in dilatometer data acquisition during Gleeble simulation. On the other hand, the dilatation measurements (Figure 3c,d) made during Q&P treatments at 250 and 200 °C showed expansions far greater than those predicted for carbon partitioning alone, revealing that the austenite-to-bainite transformation occurs during holding for 1 h (ignoring the possibility of isothermal martensite formation), thus indicating the conventional time–temperature dependence of bainite transformation, though influenced by the presence of prior martensitic laths. As Gong et al. [37] reported, the prior martensite present in the microstructure is hermal martensite phase following initial quenching to 250 and 200 °C (20 and 57 vol.%, rean accelerating parameter for subsequent bainite transformation. So, the presence of an atspectively) should have also accelerated bainitic transformation, thus shortening the incubation period and finally accelerating the bainite transformation rate, too [38,39,40]. The width of the specimen increases with time (region I) and then more rapidly on a logarithmic scale (actually at a decreased transformation rate in absolute scale) at about 60 s and 200 s for holding at 250 and 200 °C, respectively, as shown by the arrows (region II). During cooling to a particular temperature below M_s_, carbon partitioning begins immediately and continues thereafter (region I), with the possible formation and migration of ledges leading the martensite laths to grow into the austenite. During long holding, some austenite pools can transform to ultrafine lower bainite (region II). In the literature, a third region (region III) has also been reported, in which the sample contracts because of extensive martensite tempering [29,41].

By comparing the dilatation curves in Figure 3 regardless of experimental type, i.e., Q&B or Q&P process, it is obvious that the incubation time to start the bainite transformation increased continuously as isothermal holding temperature decreased from 450 °C to 200 °C. The bainite fractions at different isothermal holding temperatures estimated from the dilatation curves in Figure 3a,c are reproduced in Figure 3b,d, respectively, as a function of holding time in linear absolute scale. In Figure 3b, the transformation rate increased as holding temperatures increased between 300 and 450 °C. Figure 3d reveals that the volume fraction of the bainite phase decreased as the holding temperature below the M_s_ decreased from 250 to 200 °C, and this is due to the formation of significant initial martensite before the start of isothermal holding. A large amount of initial athermal martensite renders a significantly lower amount of untransformed austenite prior to the start of bainite transformation and hence has a low driving force as well at a relatively low holding temperature of 200 °C. A detailed account is presented in our previous work [5].

### 3.2. Kinetic Data of Bainite Phase Transformation

The linear regression equations describing the dilatation behaviour corresponding to each isothermal holding temperature plotted as ln [−ln (1 − x)] vs. lnt in accord with Avrami-type functions (Equation (3)), both above and below the M_s_ temperature, are presented in Figure 4 and Figure 5, respectively. The linear regression equations corresponding to all isothermal holding temperatures between 450 and 200 °C and the associated values of k and n are tabulated in Table 1 and Table 2, respectively. According to Table 2, a systematic change in n and k values occurred, as the temperature decreased below the M_s_. For better understanding, the variations of n and k as a function of isothermal holding temperature are displayed in Figure 6. The value of n varied between 1.0 and 1.8 for the isothermal Q&B treatments (above M_s_) and less than 0.4 for Q&P treatments (below M_s_) [31]. Referring to Figure 6, above M_s_ temperature, the Avrami exponent n increased continuously from 1.0 to 1.8 with the bainitizing temperature (Table 2), though the temperature dependent rate coefficient k varied only in a narrow range (0.002–0.004 s^−1^), showing its weak dependence in the bainitic regime. On the other hand, below M_s_ temperature, the rate coefficient k corresponding to bainite reaction for Q&P samples (0.03–0.04 s^−1^) was more than 10 times greater than that seen for Q&B samples (Figure 6), though the Avrami exponent n dropped significantly, showing a weak dependence on temperature. The decrease in the Avrami exponent during isothermal holding below the M_s_ temperature indicates that bainite formation became site-saturated because of the presence of numerous nucleation sites at prior austenite–martensite interfaces. Additionally, the lower values of k for Q&B samples (0.0019–0.0040) compared to those of Q&P specimens (0.392–0.283) (Table 2) are consistent with those reported by other research groups [23,42]. According to the investigation carried out by Zhou et al. [43], a high k value signifies a faster growth process of isothermal bainite formation. 

### 3.3. Rate of Bainite Phase Transformation

The dilatation rate curves of various specimens held above and below the M_s_, derived from the dilatation test results, are shown in Figure 7. In the case of Q&B specimens, with isothermal hold above the M_s_ temperature (Figure 7a), a peak in the dilatation rate (0.54 μm s^−1^) corresponding to the maximum transformation rate is reached in the shortest time (10 s) for the specimen isothermally held at 450 °C. As can be seen in Figure 7, a further decrease in holding temperature results in a drop in maximum transformation rate with a consequent increase in the time taken to reach the peak. In sharp contrast, for Q&P specimens held at temperatures below M_s_ temperature (Figure 7b), the maximum dilatation rate occurred at the very start of holding owing to the occurrence of various mechanisms, including partitioning of carbon and isothermal martensite formation. A high transformation rate below M_s_ is due to the presence of prior martensite, which is supersaturated from carbon, thus leading to rapid carbon partitioning from martensite to the adjacent untransformed austenite. On the other hand, the second increase in transformation rate during continuous holding relates to the bainite formation that can occur even at these low temperatures below M_s_. This is due to the performed initial martensite enabling more heterogeneous nucleation sites for bainitic transformation and also promoting a higher fraction of dislocations in the remaining austenite (dislocation structures are favorable sites for bainite formation) [44]. In addition, the formation of initial martensite increases the interior stress in the remaining austenite and so prepares additional mechanical driving force for isothermal bainitic transformation [37,45].

### 3.4. Microstructural Features and Hardness Measurements

Typical examples of microstructures of Q&B samples subjected to various isothermal holding treatments between 450 and 300 °C (above the M_s_ temperature) for 1 h, as prepared using laser scanning microscopy, are shown in Figure 8. Referring to Figure 8a, the presence of bainitic ferrite (relatively large bright regions) with a distribution of carbides (grey, small particles) between the sheaves shows that after isothermal holding at 450 °C for 1 h, the microstructure essentially consisted of bainite–carbide aggregates typical of upper bainite microstructure. A decrease in isothermal bainitizing temperature to 400 °C (Figure 8b) resulted in the refinement of the transformed phase in comparison to that at 450 °C, though a fine distribution of carbides between the sheaves still persisted. A further lowering of bainitizing temperature at 350 and 300 °C, however, resulted in a significant fraction of extremely fine bainitic laths (grey regions) after 1 h holding, as depicted in Figure 8c,d, respectively. Carbon-enriched retained austenite–martensite (RA/M) islands (bright) can be seen distributed throughout the bainitic matrix. Carbides formation, if any, was not discernible from the laser scanning micrographs, Figure 8c,d.

So far as 1 h holding in the Q&P regime at 250 °C is concerned, the structure is a mixture of martensitic packets in association with an ultrafine distribution of bainitic laths (Figure 9a). As stated above, carbon partitioning began immediately below M_s_ (region I), with possible ledged growth of martensite laths into adjacent austenite isothermally, though this could not be detected in the microstructure and needs further investigation. With continued holding, some austenite pools transformed to ultrafine lower bainite (region II). It is obvious that the interstitial carbon atoms preferentially diffuse to prior austenite soon after quenching below the M_s_ temperature [46,47,48]. The octahedral interstitial positions in face-centered cubic (FCC) austenite are larger in comparison to those in body-centered tetragonal (BCT) martensite, resulting in swift carbon partitioning from the carbon-supersaturated martensite to the untransformed austenite. Carbon content enriching the untransformed austenite causes a further decrease in bainite transformation temperature, even though the bainite transformation rate increased significantly due to the presence of prior martensitic laths. By further decreasing the holding temperature to 200 °C, a significant amount (~57%) of athermal martensite laths already formed in the microstructure and only a tiny fraction of ultrafine (nanostructured) bainite formed following holding for 1 h (Figure 9b), as the bainite formation kinetics at this low temperature were quite slow despite the presence of a high martensite phase fraction in the microstructure (see Figure 3d). Carbides formation, if any, was not discernible in this sample, similar to a specimen held at 250 °C. 

For further investigation into the presence of the RA phase and also the possibility of carbide formation, select samples from different heat-treatment conditions were examined in a TEM. Typical examples of microstructures for specimens held at different temperatures both above (350 °C) and below (200 °C) the M_s_ after 1 h isothermal holding are shown in Figure 10. A bright field (BF) image of the specimen held at 350 °C (Figure 10a) revealed that the specimen displayed bainitic laths with a high density of dislocations. The TEM observations, including dark field (DF) imaging (Figure 10b) and selected area diffraction pattern (SAED) analysis, revealed the presence of finely divided interlath films of the RA phase in the Q&B samples held at 350 °C for 1 h. By decreasing the holding temperature below the M_s_ (200 °C), the TEM image clearly depicted highly dislocated martensite and fine bainitic laths in the microstructure (Figure 10c). The SAED pattern analysis (inset in BF image; Figure 10c) and DF imaging (Figure 10d) of the relevant spots in SAED patterns appeared to show the presence of a finely divided austenite phase a few tens of nanometers in width. These results seemed to suggest that carbon atom diffusion in the austenite phase at such low temperatures happens in less than about 10 nm after 1 h isothermal holding, so a significant part of the core of the untransformed austenite is not adequately enriched with carbon to be stabilized at RT during final cooling. 

Furthermore, the presence of carbides for different processing conditions at 350 and 200 °C were investigated by TEM examination and the typical examples of TEM images are shown in Figure 11. Carbides in the samples held for 1 h above (Figure 11a,b) and below (Figure 11c,d) the M_s_ temperature can be identified with DF imaging (Figure 11b,d), using carbide diffraction spots in the concerned SAED patterns (shown as insets in the corresponding BF images presented in Figure 11a,c, respectively). The evidence of carbide formation suggests that alloying with a high Si content can delay carbide formation but cannot completely prevent it. The results confirmed that despite the low temperature of 200 °C, there was still a strong driving force for carbides formation during 1 h holding. DF images presented in Figure 11b,d show fine precipitation of interlath carbides at the nanometer scale in both Q&B and Q&P conditions.

Vickers hardness measurements were conducted on all the Q&B and Q&P samples and the data are presented in Table 3. As isothermal holding temperature decreased from 450 to 200 °C, macrohardness increased continually from 350 to 655 HV30, corroborating the notable microstructural evolution, right from the formation of upper bainite at 450 °C (Figure 8a) to extremely refined microstructures at lower temperature Q&B treatments (400–300 °C; Figure 8b–d) and subsequent multiphase microstructures realized in specimens held at 250 and 200 °C (Figure 9a,b, respectively), comprising ultrafine (nanostructured) martensite, bainite and finely divided retained austenite phase constituents. This is because the rate of carbon diffusion was significantly higher at high isothermal holding temperatures. Accordingly, the bainite transformation reaction rate was accelerated due to the faster diffusion of carbon at high temperatures for Q&B samples. The increase in hardness with the lowering of the isothermal bainitizing temperature just above the M_s_ temperature is attributed to the realization of extensively refined ultrafine bainite (Figure 8c–d, for instance). The high hardness of Q&P samples is due to the presence of a significant prior martensite fraction, in addition to bainite and/or untempered, high-carbon martensite formation during final cooling.

### 3.5. EBSD Imaging and XRD Confirmations

Examples of EBSD images, which are combinations of image quality (IQ) and phase (PM) maps of heat-treated specimens, are shown in Figure 12 and Figure 13. These EBSD images can provide much more information about the location, distribution and morphological features of different phase constituents in multiphase microstructures, though they are limited by the resolution of the technique (about 0.08 μm). Samples held at 450 °C for 1 h (Figure 12a) displayed a microstructure comprising ferritic bainite (red color) and a very small amount of RA (green color). By decreasing the isothermal holding temperature in the range of 400–300 °C (Figure 12b–d), the bainite phase (red color) becomes finer and the presence of small amounts of finely divided retained austenite phase in the bainitic matrix is also discernible, though some fractions of very fine austenite (<0.08 μm) cannot be identified and can be detected only by XRD. Furthermore, a comparison of EBSD images shows that with a decrease in the temperature from 450 to 300 °C, the volume fractions of retained austenite increased slightly from 2 to 5 vol.%. This can be explained with the theory of T_0_ curve which suggests that in high-carbon bainitic steels more supercooled untransformed austenite will be available at the end of isothermal treatment due to the increased incubation time delaying the start of bainitic transformation [26]. By decreasing the holding temperature below M_s_ temperature to 250 and 200 °C and holding for 1 h (Figure 13a,b), the EBSD images revealed the formation of multiphase microstructures consisting of ultrafine bainite, martensite and RA phases, but with a much finer division in comparison to those seen in Q&B specimens (Figure 12). These EBSD images also show that the bainitic areas are revealed as bright red regions because of a higher confidence indexing of bainitic ferrite laths, while fresh martensitic regions have relatively poor confidence indexing due to a higher intensity of internal strains and dislocations. These EBSD images confirm that by decreasing the isothermal holding temperature below the M_s_ temperature, bainite formation could not be completed and both tempered and fresh martensite phases were present in the final micrographs, in agreement with the laser scanning microscopy observations (Figure 13). 

Typical X-ray diffraction (XRD) spectra recorded on differently heat-treated samples both from Q&B and Q&P experiments are presented in Figure 14. The XRD patterns presented in Figure 14 show that, with a decrease in holding temperature from 450 to 200 °C, the peaks that relate to the FCC retained austenite phase are clearly revealed in combination with BCC ferrite peaks. These XRD patterns also corroborate the microstructures presented in Figure 12 and Figure 13, confirming the stabilization of increased retained austenite at room temperature as the holding temperature decreased from 450 to 200 °C. The RA contents were calculated using the XRD analyses and the determination of average carbon contents in RA fractions was based on the lattice parameter calculations using equation 1 and the results are presented in Table 4. As revealed in the table, by decreasing the holding temperature from 450 to 200 °C, the RA volume fraction rose significantly from ˂3 to 17.2%. As mentioned previously, during heat treatment below the M_s_ (250 and 200 °C), the nucleation sites for bainitic transformation increased due to the presence of martensitic laths and consequently increased the phase fraction of the bainite phase. A high bainite fraction may enhance the shear resistance in the austenite–bainite interface regions, which can suppress the displacive growth of bainite and finally increase the stability of austenite against bainitic transformation [23]. According to the results shown in Table 4, the carbon content of the RA phase changed in a narrow range between 1.39 and 1.45% as the temperature dropped from 350 to 300 °C (above the M_s_ temperature), but dropped later to about 1.20 wt.% at both temperatures below the M_s_ (250 and 200 °C) as a result of lower temperature partitioning of carbon atoms from bainite (and also athermal martensite) to the untransformed austenite during isothermal holding [3]. Unfortunately, the XRD measurements were not appropriate to determine the carbon contents of the Q&B specimens held at the high partitioning temperatures of 400 and 450 °C, as the RA contents were very low (<3%). Carbide formation at temperatures as low as 200 °C, as observed in TEM images (Figure 11c,d), encourages further study regarding the driving force and mechanisms behind such formations and their types and morphologies. This forms the next phase of the work.

## 4. Conclusions

In this study, the characteristics and kinetics of bainite transformation behaviour during austenite phase decomposition in a high-Si, medium-carbon steel (DIN 1.5025 grade) have been investigated in a thermomechanical simulator (Gleeble 3800) after quenching and holding, both above and below the M_s_. The dilatometer measurements and microstructural investigation using various metallography techniques (e.g., EBSD, TEM, etc.) have been carried out to understand the bainite transformation characteristics coinciding with the operation of other microstructural mechanisms that may be occurring during Q&B and Q&P processing, further confirmed by XRD and hardness measurements. The conclusions can be summarized as follows:

1.Isothermal holding above the M_s_ temperature (275 °C) facilitated bainitic transformation, which was completed after 1 h isothermal holding regardless of fractions transformed. In addition, a further decrease in holding temperature below the M_s_ resulted in the formation of initial athermal martensite, which provided additional nucleation sites for the accelerated formation of bainitic sheaves and enabled carbon partitioning to the adjacent austenitic areas, thereby enhancing the bainite transformation rate. 2.Avrami-type functions suitably predicted the progress of bainite transformation at different isothermal holding temperatures as a function of elapsed times, above and below the M_s_ (275 °C). The rate coefficients k corresponding to the bainite reactions in Q&P samples (0.03–0.04 s^−1^) were more than 10 times greater than those for Q&B samples (0.002–0.004 s^−1^), which showed a weak dependence in the bainitic regime. The Avrami exponent n, on the other hand, varied in the range 1.0 and 1.8 for the isothermal Q&B treatments (above M_s_), increasing with the increasing bainitizing temperature, but was significantly lower (<0.4) for Q&P treatments. The decrease in the Avrami exponent (n) with holding below the M_s_ (250 and 200 °C) indicated that the bainite formation became site-saturated because of the presence of numerous nucleation sites at prior austenite–martensite interfaces.3.Following Q&P processing (below M_s_ at 250 and 200 °C), a maximum rate of transformation occurred at the start of isothermal holding owing to the occurrence of such mechanisms as carbon partitioning and possible ledged growth of isothermal martensite in region I. A second peak seen in region II subsequently marked the decomposition of austenite to ultrafine bainite during longer holding.4.Microstructural investigation confirmed the dilatation results and showed the extensive formation of bainitic microstructures after isothermal holding at 450 °C, whereas multiphase microstructures comprising complex martensite–bainite–retained austenite phase constituents were realized by holding at temperatures below the M_s_ temperature (200 and 250 °C). TEM images confirmed the presence of fine films of interlath RA and carbides formed during different isothermal holding treatments for 1 h both above and below the M_s_ temperature.5.Hardness tests performed on the Q&B and Q&P heat-treated specimens corroborated the results of the dilatation curves and phase transformation evolution. Vickers hardness decreased continuously with increasing isothermal temperature between 200 and 450 °C due to the microstructural evolution from a multiphase microstructure comprising martensite–bainite–retained austenite (isothermal holding at 200 °C) to relatively complete bainitic structures at the isothermal holding temperature of 450 °C.

## Figures and Tables

**Figure 1 materials-15-00539-f001:**
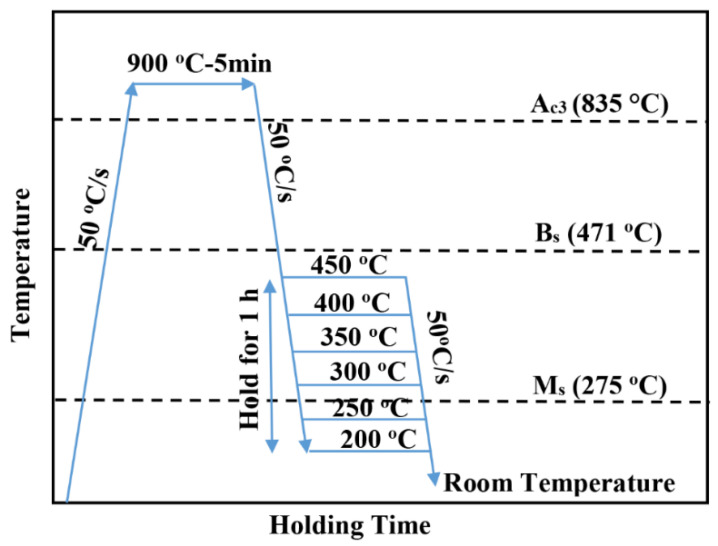
Schematic of the typical heat-treatment processes.

**Figure 2 materials-15-00539-f002:**
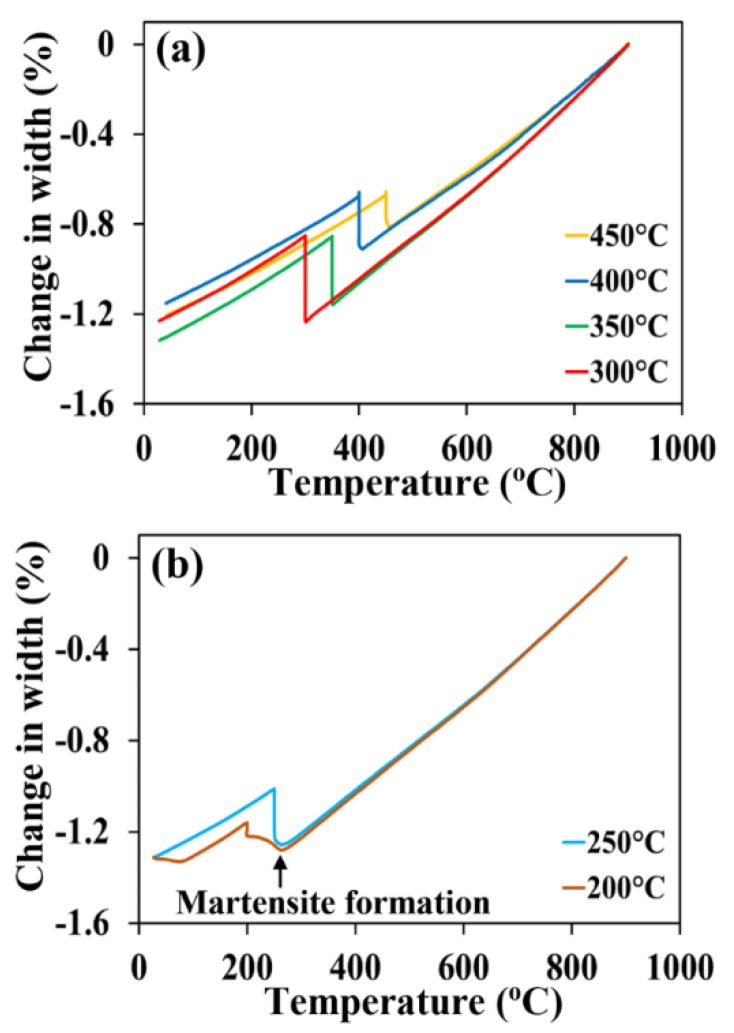
Dilatation curves reveal phase transformation occurring during cooling at 50 °C/s and holding at different temperatures and corresponding changes in specimen width: (**a**) Q&B and (**b**) Q&P treatments.

**Figure 3 materials-15-00539-f003:**
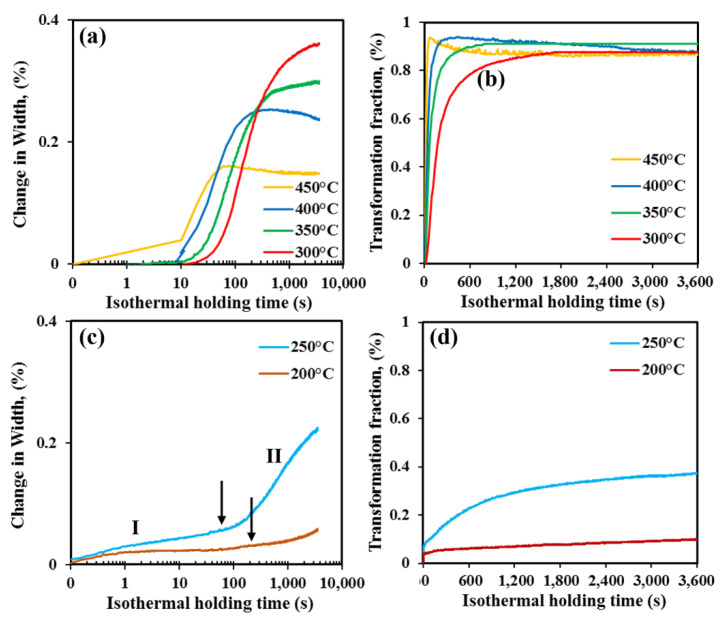
Dilatation curves showing changes in specimen width with time corresponding to isothermal holding for 1 h at different temperatures: (**a**,**b**) Q&B and (**c**,**d**) Q&P treatments shown in both logarithmic (**a**,**c**) and linear (**b**,**d**) scales.

**Figure 4 materials-15-00539-f004:**
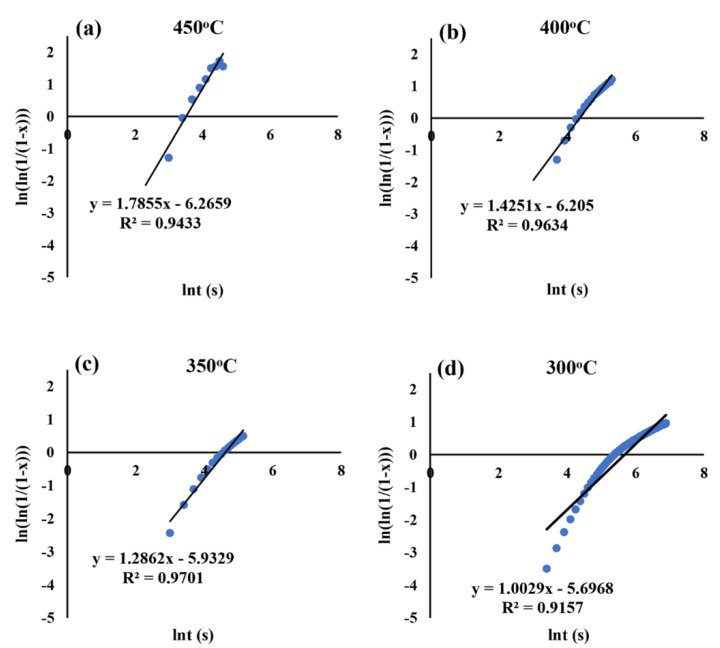
Plots of ln[−ln(1−x)] vs. lnt corresponding to dilatation data acquired at various isothermal holding temperatures above M_s_: (**a**) 450 °C, (**b**) 400 °C, (**c**) 350 °C and (**d**) 300 °C.

**Figure 5 materials-15-00539-f005:**
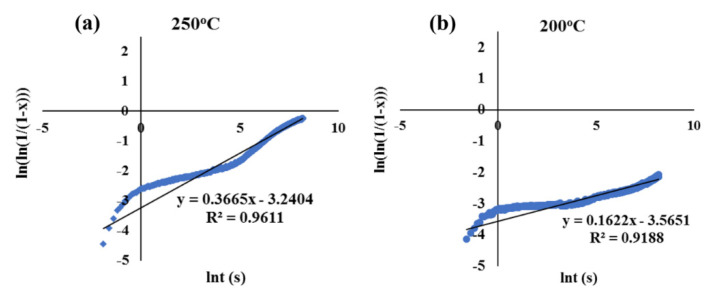
Plots of ln[−ln(1 − x)] vs. lnt corresponding to dilatation data acquisitioned at various isothermal holding temperatures below M_s_: (**a**) 250 °C and (**b**) 200 °C.

**Figure 6 materials-15-00539-f006:**
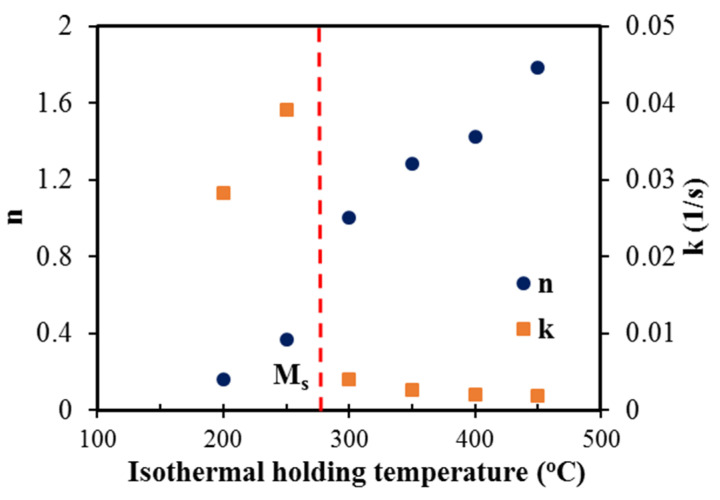
The variations of n and k with the isothermal holding temperature between 450 and 200 °C.

**Figure 7 materials-15-00539-f007:**
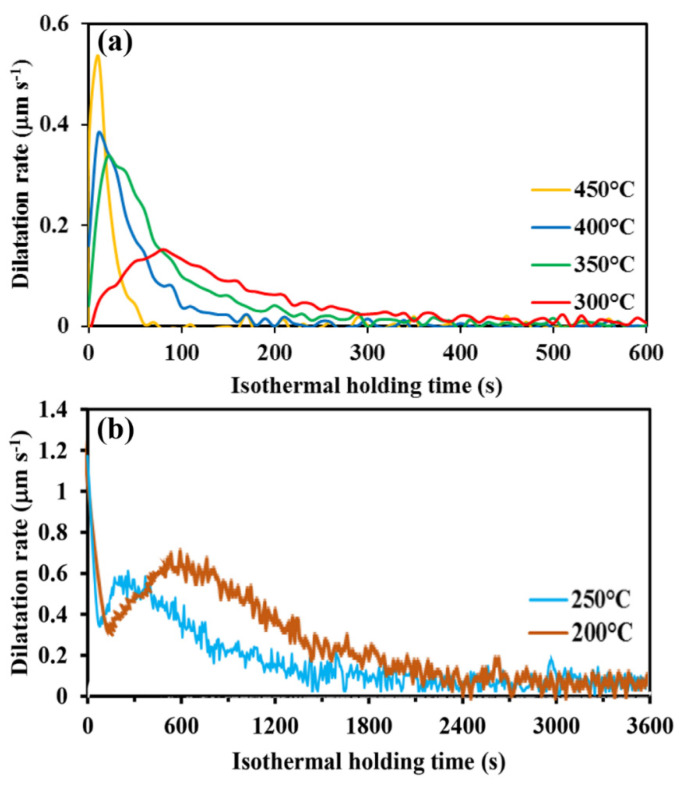
Dilation rate curves as a function of holding time during treatment in (**a**) Q&B and (**b**) Q&P conditions.

**Figure 8 materials-15-00539-f008:**
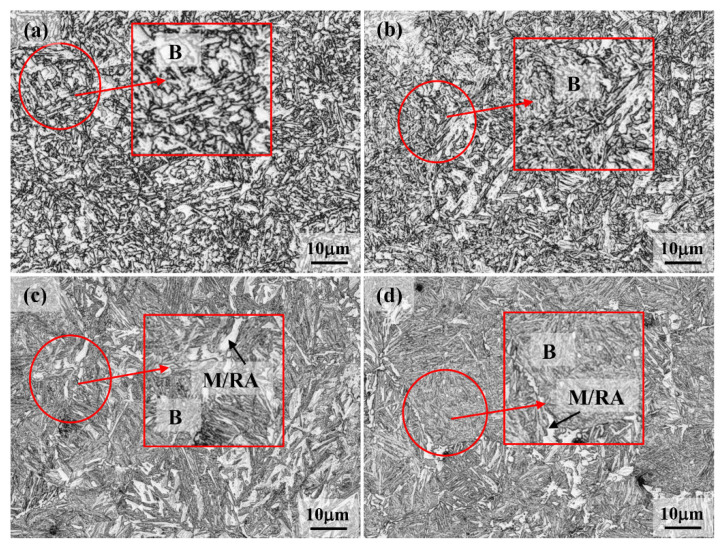
Micrographs recorded on Q&B samples quenched at (**a**) 450 °C, (**b**) 400 °C, (**c**) 350 °C and (**d**) 300 °C temperatures and held for 1 h isothermal holding. B and M/RA represent bainite and martensite–retained austenite islands, respectively.

**Figure 9 materials-15-00539-f009:**
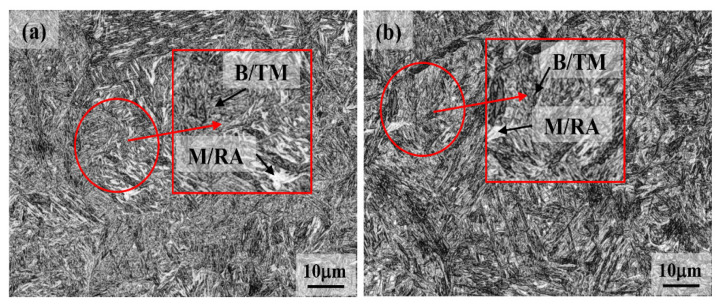
Micrographs recorded on Q&P samples were quenched at (**a**) 250 °C and (**b**) 200 °C and held for 1 h isothermal holding time. TM, B and M/RA represent tempered martensite, bainite and martensite–retained austenite islands, respectively.

**Figure 10 materials-15-00539-f010:**
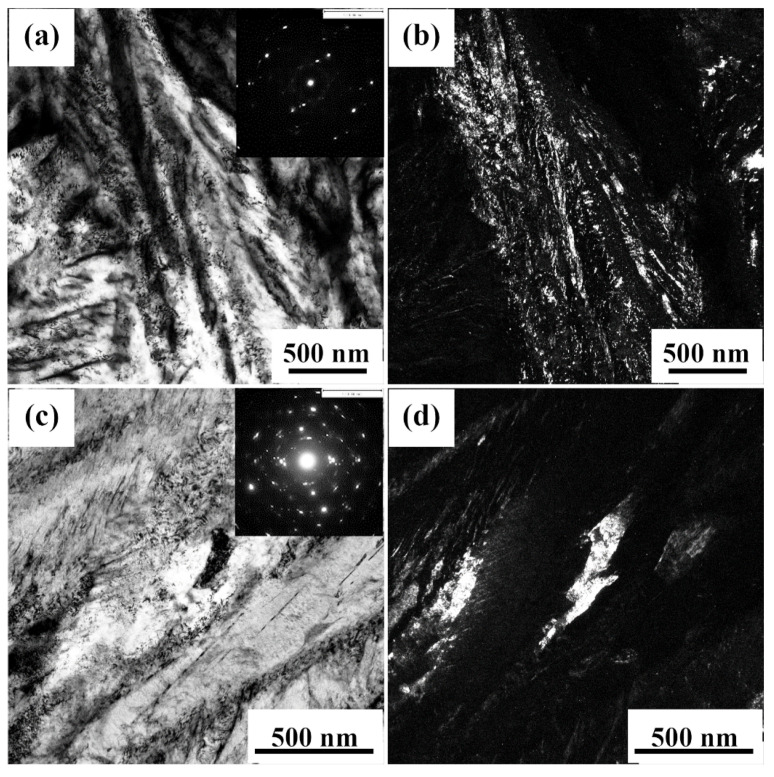
TEM images of specimens held at 350 °C (**a**,**b**) and 200 °C (**c**,**d**)) for 1 h. The BF images with inset SAED patterns and the DF images from the diffraction spots are shown for RA phase detection.

**Figure 11 materials-15-00539-f011:**
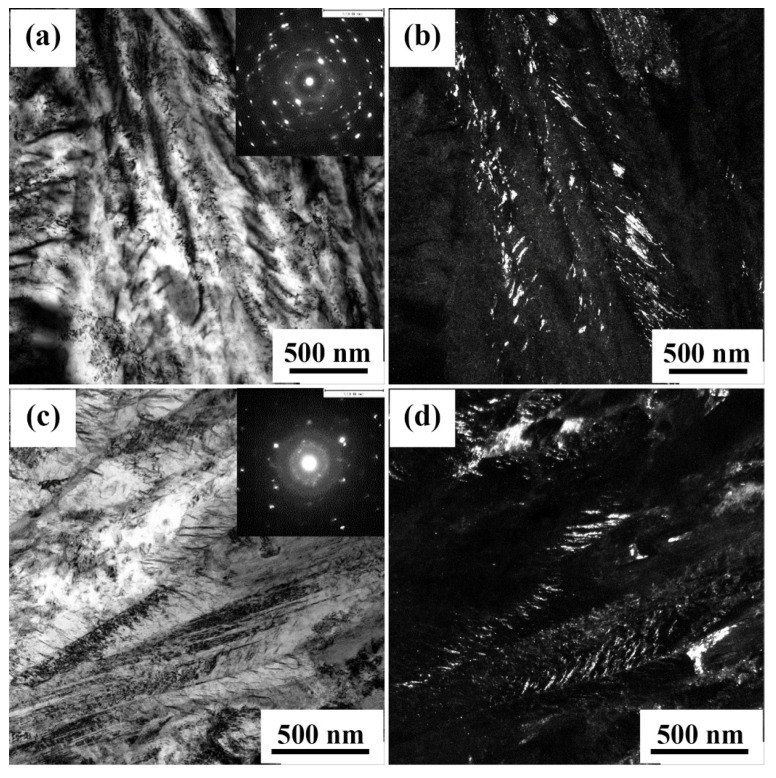
TEM images of specimens held at 350 °C (**a**,**b**) and 200 °C (**c**,**d**) for 1 h. The BF images with inset SAED patterns and the DF images from the diffraction spots are shown for carbide detection.

**Figure 12 materials-15-00539-f012:**
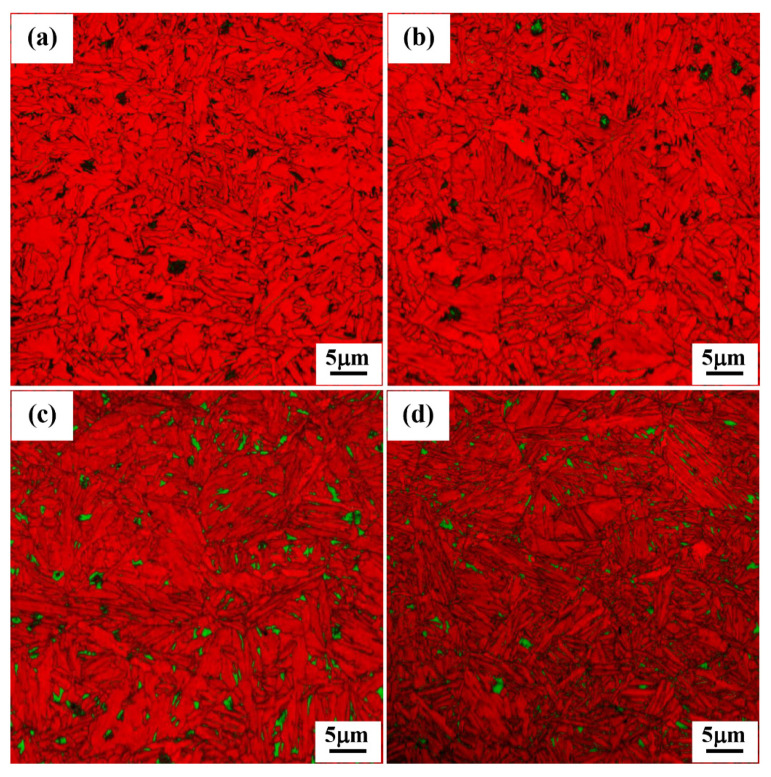
EBSD images of samples held at: (**a**) 450 °C, (**b**) 400 °C, (**c**) 350 °C and (**d**) 300 °C for 1 h isothermal holding. The RA is shown in green and the bainite is red-colored.

**Figure 13 materials-15-00539-f013:**
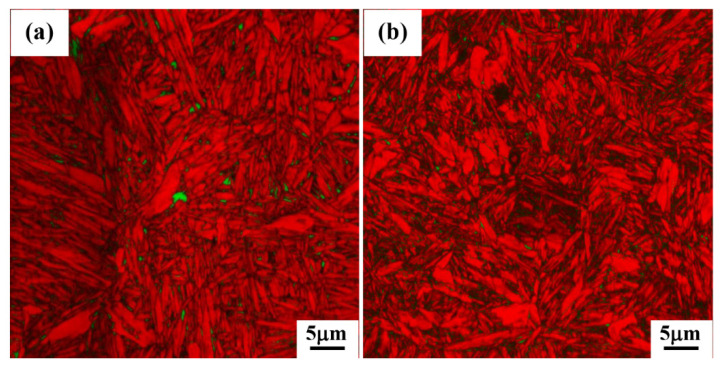
EBSD images of specimens held at (**a**) 250 °C and (**b**) 200 °C for 1 h isothermal holding. The RA is shown in green and the bainite and martensite are the bright and dark red-colored regions, respectively.

**Figure 14 materials-15-00539-f014:**
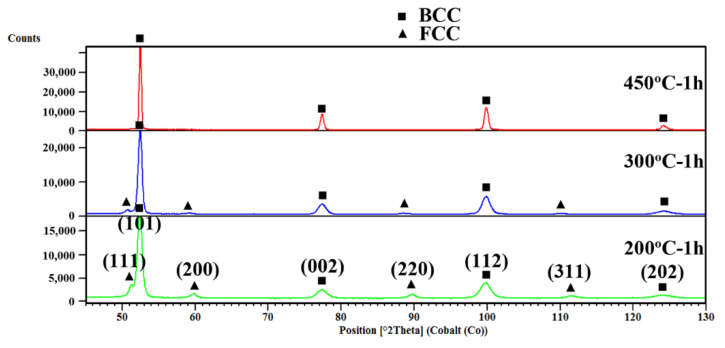
XRD pattern recorded for both the Q&B and Q&P specimens at different temperatures between 450 and 200 °C for 1 h isothermal holding.

**Table 1 materials-15-00539-t001:** Linear regression equations describing the dilatation behaviours at various isothermal holding temperatures based on Avrami-type functions applied to the dilatation results in the temperature range 450–200 °C.

Isothermal Holding Temperature (°C)	Linear Regression Equations	
450 °C	y = 1.7855x − 6.2659	R^2^ = 0.9433
400 °C	y = 1.4251x − 6.205	R^2^ = 0.9634
350 °C	y = 1.2862x − 5.9329	R^2^ = 0.9701
300 °C	y = 1.0029x − 5.6968	R^2^ = 0.9157
250 °C	y = 0.3665x − 3.2404	R^2^ = 0.9611
200 °C	y = 0.1622x − 3.5651	R^2^ = 0.9188

**Table 2 materials-15-00539-t002:** Parameters n and k derived from linear regression equations describing the dilatation behaviours at various isothermal holding temperatures in the range 450–200 °C.

Isothermal Holding Temperature (°C)	n	k (1/s)
450 °C	1.7855	0.0019
400 °C	1.4251	0.0021
350 °C	1.2862	0.0027
300 °C	1.0029	0.0040
250 °C	0.3665	0.0392
200 °C	0.1622	0.0283

**Table 3 materials-15-00539-t003:** Macrohardness measurements (HV30) of Q&B and Q&P specimens.

Isothermal Holding Temperature (°C)	Macrohardness (HV30)
450 °C	350
400 °C	430
350 °C	530
300 °C	584
250 °C	600
200 °C	655

**Table 4 materials-15-00539-t004:** The average RA phase fractions and their carbon contents in the case of Q&B (450, 400, 350, 300 °C) and Q&P (250, 200 °C) specimens held for 1 h.

Isothermal Holding Temperature (°C)	RA (vol.%)	Carbon Content of RA (wt.%)
450	˂3	-
400	˂3	-
350	6.9	1.39
300	6	1.45
250	7.8	1.21
200	17.2	1.20

## Data Availability

MDPI Research Data Policies at https://doi.org/10.17632/3w5rnpkcmw.1, accessed on 6 November 2021.

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
