# Peer review of "Characteristics and Kinetics of Bainite Transformation Behaviour in a High-Silicon Medium-Carbon Steel above and below the Ms Temperature"

_materials, 2022, doi:10.3390/ma15020539_

Round 1
Reviewer 1 Report
The present work investigates the effects of temperature on bainite transformation kinetics and microstructure. It can give references for designing bainite processes. The results show that bainite transformation blew Ms can be divided into two regions due to martensite-promoted and bainite-promoted. Also kinetics models were analyzed by the authors. However, some major revisions should be considered before considering publication.
- LSCM was usually termed as high temperature in situ observation in most literatures. In the present work, it was just OM. To avoid misunderstanding, please delete it.
- 20% and 57% martensite in sample 250 c and 200 c, how to measure martensite fractions? By dilation, or combined dilation and XRD? That is important to clear the subsequent transformation fractions and rate of bainite.
- In Eq.(1) Xs should be Xsi.
- lines 193-194, actually, sample 450 and 400 are matched, sample 350 and 300 are matched, so I do not think that the scatter of the cooling curves is due to data acquisition. Please recheck the cooling rate in each case, dilatation versus time figures can be given there.
- Line 239-240, “On a logarithmic scale, the sample width increases 239 slowly (region I) and then more rapidly at about 60 s and 200 s for isothermal holding at 240 250 and 200 °C, respectively”, this description should be corrected. It is obvious from Figure 3b that transformation rate was rapidly at the beginning due to prior martensite. The author ignored the influence of time, instead of visual misunderstanding.
- In Figure 3, the transformation fractions should be clearly defined (absolute or relative value). Microstructural results show that martensite and austenite are existed, thus absolute 100% bainite is not possible. Please note the accordance of Figures 3b and d.
- In Figure 3a and Figure 2a, the width change of sample 300 c is not corresponded. Figure 2 shows that 300 c has the largest dilation during isothermal holding.
- Lines 249, “it is clear that the incubation time to start the bainite transformation increased continuously with the decrease in isothermal holding temperature right 250 from 450 °C down to 200°C”, then how to indicate that prior martensite accelerate bainite transformation. This description was not cautious and contradicted with Figured 3d, where the transformation rate is actually larger at the beginning.
- Conclusions must be simplified. 3~4 points are reasonable. Delete the unimportant ones.
Author Response
"Please see the attachment."

Reviewer 2 Report
In this paper, the bainite transformation behavior of low alloy, high-silicon DIN 1.5025 steel was investigated. The influence of isothermal holding temperature on bainite transformation kinetics and microstructure characteristics was analyzed by experiments and calculations, which has a good guiding significance for the microstructure control, establishing the relationship between microstructure and mechanical properties and deepening the theory under this composition system. Main comments are shown as follows:
- In the part of abstract, the first sentence and the second sentence are slightly repetitive, which mainly express the kinetic of microstructure transformation during isothermal holding above and below Ms, and suggest that the language should be organized reasonably to avoid repetitive expression;
- At the end of the introduction, it is necessary to point out the significance and importance of the study, in addition to ‘the study of bainite transformation kinetics in low-alloy, high-silicon medium carbon steels is not yet mature’;
- In the result part, pay attention to the sequence labeling of each subsection and keep the font correct according to the format requirements, starting from 3.1 followed by 3.2 and 3.3 and so on;
- In the third part of the results, when the temperature is lower than Ms, the maximum transformation rate is reached at the beginning of isothermal holding, and the subsequent curves show a trend of increasing first and then decreasing. It is suggested to explain in detail why the transformation rate of bainite show this rule.
Author Response
"Please see the attachment."

Round 2
Reviewer 1 Report
The paper was well revised, thus I suggest it to be accepted.